# Dissecting learning and forgetting in language model finetuning

**Xiao Zhang & Ji Wu**
Department of Electronics Engineering
Tsinghua University
`xzhang19@mails.tsinghua.edu.cn,wuji_ee@mail.tsinghua.edu.cn`

## Abstract

Finetuning language models on domain-specific corpus is a common approach to enhance their domain knowledge and capability. While improving performance on domain tasks, it often brings a side-effect of forgetting of the model's general abilities. In this study, we analyze the effects of finetuning on language models by dissecting its impacts on the modeling of topic, style, and factual knowledge in text. Our method uses instruction-following LLMs such as ChatGPT to auto-generate controlled-variable text examples which we use to probe the model. Our findings reveal that finetuning results in significant shifts in the language model's topic and style priors, while actual knowledge learning only contributes to a small fraction of the total probability change. Analysis shows that the adaptation of topic and style priors behave akin to learning simple features: they are learned rapidly and require little model capacity. They are also learned independently and primarily at the beginning of a text sequence. In contrast, factual knowledge is learned stably but slowly and requires significant model capacity. The findings offer insights and understanding into the finer dynamics of learning and forgetting in language models, and potentially inform future research on improving domain adaptation and addressing the challenges of continual language learning.

## 1 Introduction

Large language models (LLMs) pre-trained on general corpus show impressive common-sense knowledge, reasoning ability, and zero-shot performance on a variety of tasks (OpenAI, 2023; Touvron et al., 2023; Chung et al., 2022). Finetuning LLMs on domain corpus further enhances their domain knowledge and ability, substantially improving performance on domain tasks (Lewkowycz et al., 2022; Chen et al., 2021; Singhal et al., 2023). However, it is also observed that finetuning language models can lead to forgetting of previously learned information (Jang et al., 2022; Chen et al., 2020), which is often mitigated in practice by mixing general corpus with domain data in finetuning (Rozière et al., 2023; Ouyang et al., 2022).

To better understand the effect of finetuning on a language model (specifically, we study "domain finetuning"of general models on a domain corpus), we perform a dissection analysis on how language models model different factors of text. We analysis the topic (the overall theme, e.g., "language model finetuning"), the style (the structure, tone and diction, e.g., academic writing in ICLR paper format), and the factual knowledge (detailed factual information, e.g., methods, citation, and results in this paper) as three main components of text. As language models represent probability distributions of text, this dissection allows us to observe how they assign probabilities to text of different content during finetuning. This gives a finer and alternative perspective compared to existing analysis based mainly on downstream task performance.

To understand the behavior of language models, a common approach is probing language models with specifically designed examples (Srivastava et al., 2022; Lin et al., 2022). We create samples of text with specific combinations of content and style and use them to query the language model's likelihood. We study open LLMs such as LLaMA (Touvron et al., 2023) domain finetuned with the conventional causal language modeling recipe used in pre-training. Following a recent trend of automatic data generation with LLMs (Honovich et al., 2023; Ho et al., 2023), we use ChatGPT to

systematically generate high-quality text samples, enabling controlled-variable probing of language models with minimal human effort in data curation.

Our investigation reveals that while domain finetuning enhances domain knowledge, it also induces strong topic and style biases in the language model towards the training data, making the model much less likely to generate text with other topics and styles. More interestingly, we found many characteristics that differentiate the learning dynamics of simple topic and style biases vs. factual knowledge. The following two findings summarizes the main contributions of our analysis:

- **Domain finetuning leads to a significant change in the topic and style priors of the language model, biasing them towards the training data. Effect caused by such bias dominates the learning and forgetting observed in finetuning**. The learning of factuals knowledge only contributes to a small part of the change in modeling probabilities, which offers a possible explanation of the difficulty in preserving general abilities while assimilating knowledge in domain finetuning.

- **Topic and style biases are learned like simple features, while factual knowledge are learned like complex features in finetuning**. Biases are learned rapidly with a strength growing with the learning rate, and they require little model capacity to learn. Even considerable dataset debiasing only partially mitigates them. The biases are also predominantly acquired at the beginning of the text sequence, independent of other biases. In contrast, factual knowledge is learned stably, relatively unaffected by token position, learning rate, or data mixture. The learning of factual knowledge also requires significant model capacity available for finetuning.

Our finding suggests that domain finetuning of language models has potential for improvement in the light of a better understanding of the learning dynamics. They could also help us identify the sources of catastrophic forgetting (French, 1999) in language models in order to facilitate effective lifelong learning of general purpose LLMs. Our data [1] and code [2] are made publicly available.

## 2 METHOD

### 2.1 ESTIMATING CONTENT AND STYLE PROBABILITIES IN A LANGUAGE MODEL

Our method involves estimating content and style probabilities under a language model by querying it with specific text examples. With a generative model $p$ of text, we can roughly decompose the probability of a document $x$ into its generating factors. In this study, we assume that $x$ is mainly determined by three factors: topic (the main topic of text), style (the writing style), and factual (the factual knowledge included in the text):

$$
\begin{aligned}
p(x) &= p(\text{topic}, \text{style}, \text{factual}) \\
&= p(\text{topic})p(\text{factual}|\text{topic})p(\text{style}|\text{topic,factual}) \\
&= p(\text{topic})p(\text{factual}|\text{topic})p(\text{style}|\text{topic}) \quad\quad (1)
\end{aligned}
$$

Note that the decomposition is only approximate and may not reflect the true generating process of text. The factors, their granularity, and the order of dependence are chosen for convenience of the analysis of the particular factor we are interested in. To simplify the analysis, we make a reasonable assumption that the factual and style are independent given the topic.

Suppose we want to estimate the probability of different styles under model $p$: consider two documents $x_A$ and $x_B$ sharing an identical topic and factual content but written in styles A and B, respectively. The likelihood ratio between these documents under $p$ becomes the likelihood ratio of the two styles (conditioned on the content).

$$
\frac{p(x_A)}{p(x_B)} = \frac{p(\text{style}_A|\text{topic})}{p(\text{style}_B|\text{topic})}
$$

---

[1] https://huggingface.co/datasets/xiaozeroone/pubmed_derived*
[2] https://github.com/xiaozeroone/lm_finetune_dissect*

Now that we want to estimate the likelihood of style A vs. style B, we can use a dataset of document pairs $\{(x_{iA}, x_{iB})\}_{i=1}^{N}$, where $x_{iA}$ and $x_{iB}$ *only* differ in style. All documents also have the same topic. The likelihood ratio can be estimated by averaging over the dataset to smooth out its possible dependency on specific documents:

$$\log \frac{p(\text{style}_A | \text{topic})}{p(\text{style}_B | \text{topic})} \approx \frac{1}{N} \sum_{i=1}^{N} \log \frac{p(x_{iA})}{p(x_{iB})} \tag{2}$$

$$= \frac{1}{N} \sum_{i=1}^{N} \log p(x_{iA}) - \frac{1}{N} \sum_{i=1}^{N} \log p(x_{iB}) \tag{3}$$

which can be easily calculated for causal language models as the difference between the average cross-entropy loss on the two set of examples $\{x_{iA}\}_{i=1}^{N}$ and $\{x_{iB}\}_{i=1}^{N}$.

The likelihood ratio between various topics can be similarly estimated by changing the order of decomposition in Eq. 1. We do not get the raw probability, e.g., $p(\text{sports})$, but we can use the likelihood ratio, e.g., $p(\text{sports})/p(\text{politics})$, to learn about the topic probabilities. Though language models do not explicitly learn a topic distribution like LDA topic models (Blei et al., 2003), they could model an implicit topic variable through approximate Bayesian inference (Wang et al., 2023a).

For the factual factor, we are interested in the likelihood ratio of factual vs. counterfactuals, e.g., $p(\text{"the sky is blue"})/p(\text{"the sky is red"})$, because such ratio represents the modeling of knowledge in the model. Calculating the ratio would require pairs of documents that use factural and counterfactuals with the same topic and style.

## 2.2 MANIPULATING CONTENT AND STYLE IN TEXT WITH INSTRUCTION-FOLLOWING LLMS

Documents that differ in only one factor, e.g., style, might not be easy to find in existing corpus. We leverage the language understanding and instruction following capabilities of instruction-finetuned LLMs to rewrite existing documents, letting it identify and manipulate the content and the style of text. We found that with appropriate prompts, LLMs such as ChatGPT can generate high-quality rewrites of a passage, altering the style while preserving the content and vice versa. For example, we can explicitly ask ChatGPT to change the topic, the factual, or the style of a passage, while keeping other elements unchanged (Figure 1):

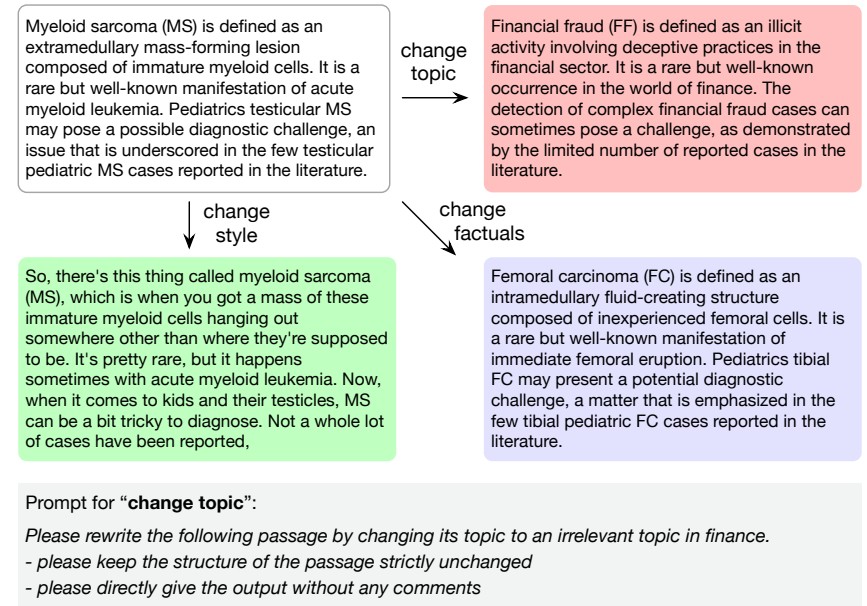

Figure 1: An illustration of changing the content and the style of a PubMed abstract with ChatGPT.

The results from rewriting show that ChatGPT effectively satisfies these strict rewriting requirements. For instance, it can produce new content compatible with the original text's structure and

style. We found that GPT-3.5 is capable enough for this task, although GPT-4 (OpenAI, 2023) produces more successful rewrites for harder cases, and its performance is less sensitive to the prompt.

Effective rewriting consists of generating good quality text and adhering to the instruction. We use language modeling perplexity as a measure of quality and naturalness and found that the ChatGPT generated rewrites typically have a low increase in perplexity which indicates their quality. Adherence is measured by human judging whether the rewrite successfully complies with the instruction. We found that ChatGPT has a high success rate of around 95% (see Appendix A.3 for evaluation).

## 3 RESULTS

### 3.1 ANALYSIS SETUP

**Data.** We utilize two corpus in our analysis: PubMed[1], a collection of biomedical papers abstracts, and C4 (Raffel et al., 2020), a large corpus of web text. PubMed is commonly used in finetuning language models for the biomedical domain (Yasunaga et al., 2022; Luo et al., 2022; Wu et al., 2023). We use it as a representative of domain corpus and use it to finetune LLMs. We use C4 as a representative of general-domain corpus and use it for evaluation.

For probing the content and style probabilities as in Section 2.1, we use ChatGPT to rewrite documents from PubMed and C4 as described in Section 2.2. To make the analysis tractable, we sample two random subsets of 1000 documents from PubMed and C4, and then rewrite the documents with ChatGPT to generate documents with their topic, style, and factual changed. The generated derived datasets are listed in Table 1. Instructions used for generating each dataset and examples from the derived datasets are listed in Appendix A.1 and A.2.

| Dataset | Source | Topic | Factuals | Style |
|---|---|---|---|---|
| *Original datasets* | | | | |
| PubMed | - | biomedical | factual | academic |
| C4 | - | nonbiomedical* | factual* | nonacademic* |
| *Derived datasets* | | | | |
| PubMed-nonbiomedical | PubMed | **nonbiomedical** | factual | academic |
| PubMed-counterfactual | PubMed | biomedical | **counterfactual** | academic |
| PubMed-casual | PubMed | biomedical | factual | **casual** |
| PubMed-rap | PubMed | biomedical | factual | **rap** |
| C4-biomedical | C4 | **biomedical** | factual* | nonacademic* |
| C4-counterfactual | C4 | nonbiomedical* | **counterfactual** | nonacademic* |
| C4-academic | C4 | nonbiomedical* | factual* | **academic** |

Table 1: Datasets used for probing language models. Derived datasets are generated from the original datasets by rewriting with ChatGPT. Bold indicates the factor that is changed from the original dataset. * means "mostly", as C4 is a general web corpus that could contain a small portion of biomedical, academic, or counterfactual text.

In the following analysis, we calculate log-likelihood ratios by subtracting the negative causal language modeling loss $l$ between a derived and an original dataset as in Equation 3. For example, to measure the likelihood of biomedical topic vs. nonbiomedical topic, we calculate

$$\log \frac{p(\text{biomedical})}{p(\text{nonbiomedical})} = l(\text{PubMed}) - l(\text{PubMed-nonbiomedical})$$

While we focus on the PubMed corpus in most parts of our analysis, we also apply the same protocol to two more domain corpus, Pile of Law Henderson et al. (2022) in the legal domain and Amazon reviews Ni et al. (2019) in the customer review domain, for comparative analysis. Results of limited experiments on those two domains are deferred to Appendix D.

**Finetuning setup.** We finetune three language models, GPT-2 XL (Radford et al., 2019), LLaMA 2 7B and LLaMA 2 13B (Touvron et al., 2023), on the PubMed abstracts using conventional causal

---

[1] https://pubmed.ncbi.nlm.nih.gov. We use the annual baseline data of 2023.

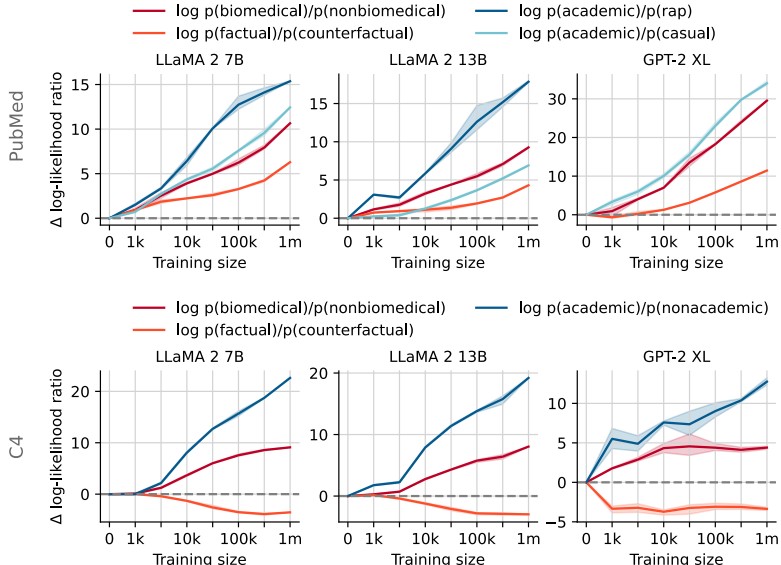

Figure 2: Change of likelihood ratios of content and style factors with the amount of training, averaged over three runs (shaded area represents max/min values). Significant bias towards the training topic and style is observed in finetuning.

language modeling loss. We finetune models on subsets of different sizes, up to 1 million abstracts. We use both full-finetuning and low-rank finetuning (Hu et al., 2022). We use AdamW optimizer (Loshchilov & Hutter, 2019) with a learning rate of 3e-6 for full-finetuning LLaMA and 1e-4 for full-finetuning GPT-2 XL and low-rank finetuning of LLaMA, all with 10% warm-up and linear learning rate decay. Learning rates are selected for each model using a grid search on a validation set. The batch size is set to 64. Other details of finetuning can be found in Appendix B.

## 3.2 The changing topic and style priors during LM finetuning

**Domain finetuning leads to significant change in topic and style probabilities.** Figure 2 shows the change of likelihood ratios between different topics, styles and factual during finetuning. The likelihood of the dominant topic (biomedical) and style (academic) in the PubMed corpus increases significantly during finetuning with respect to other topics and styles. This implies an increase in the prior probability of the training topic and style and a decrease of other topics and styles in the finetuned model. Comparing the likelihood ratio of styles academic/casual with academic/rap, it is clear that the probabilities of styles that are more different from the training style (rap) have greater reduction than styles that are closer to the training style (casual).

All the likelihood ratios change monotonically, with most showing an approximate log-linear relationship with the amount of finetuning data. The topic and style prior probabilities are continually biasing towards the finetuning data. The factual/counterfactual likelihood ratio changes at a slower rate, reflecting the learning of new factual knowledge from the domain data and the forgetting of factual knowledge in the pre-training data. We show that the factual ratio correlates well with downstream question answering performance in Appendix D.

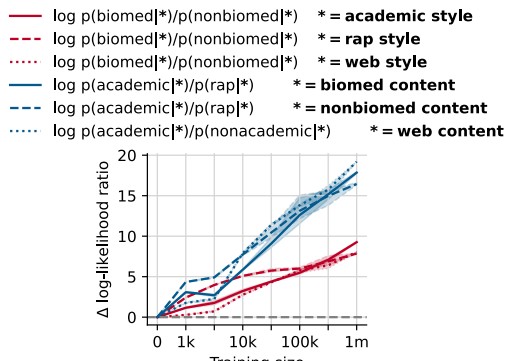

Figure 3: Likelihood ratios of topics conditioned on different styles and vice versa (LLaMA 2 13B). Likelihood ratios of topics and styles are largely independent of each other.

**Learned topic and style biases are independent.** Figure 3 shows the likelihood ratios of topics conditioned on different styles and of styles conditioned on different topics. Notably,

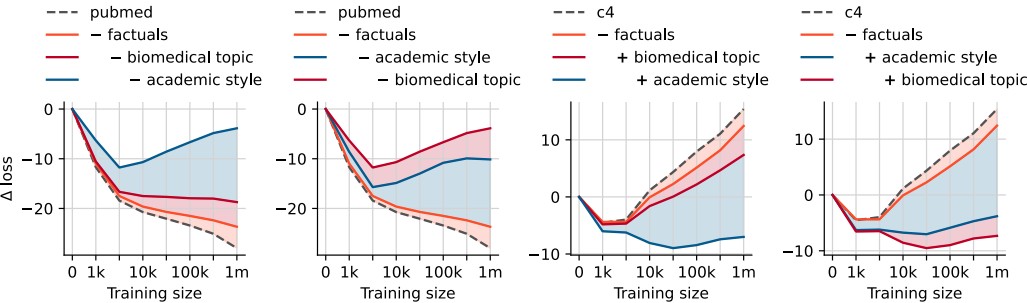

Figure 4: Ablating learning on PubMed and forgetting on C4 by evaluating on derivative datasets (LLaMA 2 13B). Colored area show the loss change introduced by each factor. "- factuals" means switching from factual to corresponding counterfactual dataset. The rest "-" and "+" mean removing or adding a style or a topic from the dataset. The graphs show that topic and style adaptation contributes to the main part of the loss change in both learning and forgetting.

the likelihood ratios of topics are changing similarly under different styles and vice versa, except for very small training sizes. This suggests that the learned topic and style biases in finetuning are generally independent of each other.

This independence would allow us to drop the conditioning in the likelihood ratios in Equation 2 and let us study the change of topic and style probabilities separately.

### 3.3 ABLATING LEARNING AND FORGETTING IN LM FINETUNING

Evaluation on derived datasets allows us to ablate the effect of learning and forgetting in language model finetuning by introducing or removing one factor at a time. Here, we use learning to refer to the loss reduction on the domain corpus and forgetting to refer to the loss increase on a general corpus (which roughly represents the data distribution in pre-training) in language model finetuning.

Figure 4 shows the ablation results, with learning measured on PubMed and forgetting measured on C4. The adaptation to biomedical topic and academic style contributes to the main part of the loss reduction on PubMed and loss increase on C4. This shows that the change of topic and style prior probabilities is the main cause of the observed learning and forgetting in language model finetuning.

However, the goal of finetuning on domain corpus is usually acquiring domain knowledge rather than adapting to the topic and style of the domain text. We can see that the learning and forgetting of factuals is steadily increasing with the amount of training data, although it only contributes to a small portion of the total loss change. This shows that adaptation to domain topic and style is a significant and probably unavoidable side effect of domain finetuning. This overly strong adaptation is one possible reason for the catastrophic forgetting observed in the finetuning of language models.

### 3.4 CHARACTERISTICS OF BIAS LEARNING AND KNOWLEDGE LEARNING

We next delve into the distinct characteristics of bias learning and factual knowledge learning during language model finetuning.

**Topic and style biases are most significant on the first few tokens and are learned quickly.** To look at more details on how each factor affects the probability of a document, we compute likelihood ratios separately for tokens at various positions within the text. Figure 5 shows that the likelihood ratios are clearly changing in different ways for the first few tokens and later tokens. The topic and style biases are much more significant at the beginning of the document (position $\leq 10$) and are quickly learned with 1-10k documents. This implies that in unconditional generation, the finetuned language model will be much more likely to generate text with the topic and style of the finetuning data by preferring those topics and styles early in generation.

For later tokens (position $\geq 100$), the topic and style biases are much weaker in comparison but are growing steadily with increased training data. The biases in later tokens seem quite consistent regardless of position till the end of text, thus contributing significant change to the whole document's probability. This part likely affects the conditional generation of language models by slightly biasing the generated text towards the topic and style of the finetuning data in each generation step.

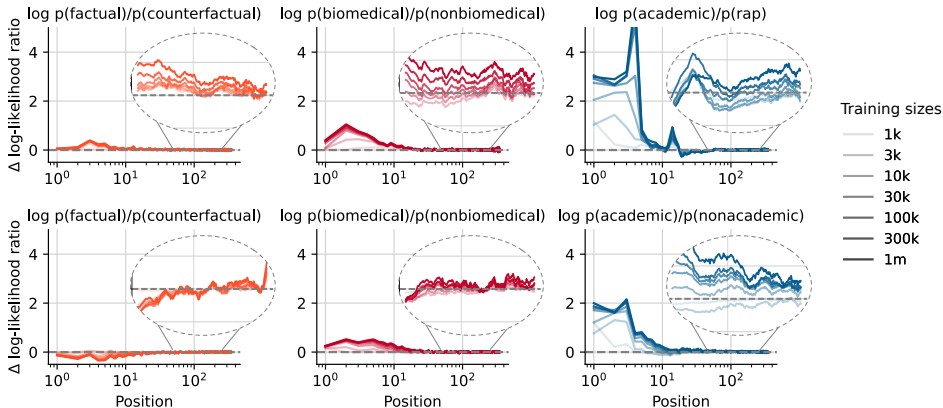

Figure 5: Likelihood ratios by tokens at different positions in the document (LLaMA 2 13B). Inside circles are zoomed-in views of the curve at around position $10^2$. Topic and style biases are most significant at the beginning of a sequence and are learned quickly with little training data.

Compared to topic and style biases, the learning of factuals appears more uniform across positions. This is because factual information can appear at any position, and there can be independent appearances of multiple factuals within one document. The difference in learning speed is likely that topic and syle are simple features that are easier to learn, and that the model may alrady seen these features during pre-training on general corpora, unlike factual knowledge which are more domain-specific.

**Topic and style biases require minimal capacity to learn, knowledge learning requires much more.** To examine the different natures of bias and knowledge learning, we finetune LLaMA 2 7B with variable numbers of trainable parameters to simulate different capacities available during finetuning. In full-finetuning, all parameters are tunable. In low-rank finetuning, only a small number of parameters are tunable, controlled by the rank $r$. For example, for $r = 8, 2$, and $1$, the number of tunable parameters is 0.3%, 0.07%, and 0.04% of the total model parameters.

Figure 6 compares the change of likelihood ratios with full and low-rank finetuning at different $r$. Interestingly, topic and style biases are learned comparably to full finetuning with just 0.02% of tunable parameters. On the other hand, factual learning is significantly hindered by low-rank finetuning at large training sizes. This suggests that the changes of topic and style probabilities are simple biases that only require adjustments in a low-dimensional subspace of the model's representations, whereas factual knowledge learning may encapsulate encoding a large number of complex patterns which requires much more model capacity.

(Side note: the learning of factuals can cause a decrease of $l(\text{PubMed})$ therefore is also affecting the topic ratio $p(\text{biomedical})/p(\text{nonbiomedical})$ on PubMed. When capacity is limited, the topic ratio and factual ratio simultaneously reduce on Pubmed in Figure 6.)

**Topic and style biases magnify with learning rate, knowledge learning does not.** We also examine the effect of learning rate on the learning of different factors. Figure 7 shows that the learned topic and style biases increase with the learning rate and are non-saturating, while the learning of factual remains consistent and does not increase with the learning rate.

This shows that a large learning rate magnifies learned bias, which is also correlated with the forgetting of general abilities (evaluated in Appendix D). A smaller learning rate might suffice for knowledge learning and offers a better tradeoff between learning new knowledge and preserving existing knowledge and abilities in domain finetuning.

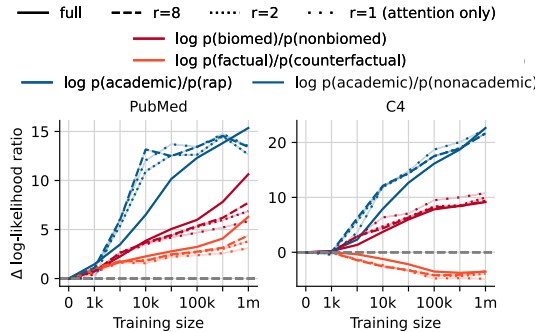

Figure 6: Likelihood ratios under different model capacities for finetuning (LLaMA 2 7B). Larger rank $r$ corresponds to more trainable parameters. Topic and style biases are learned with minimal capacity. Factual learning requires much more.

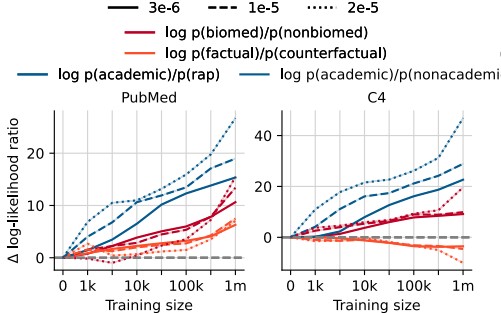 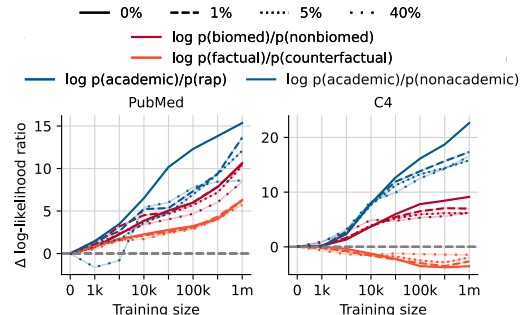

Figure 7: Likelihood ratios with different learning rates for finetuning (LLaMA 2 7B). Topic and style biases magnifies with larger learning rate, while factual learning is unaffected.

Figure 8: Likelihood ratios of training with different percentages of Wikipedia text (LLaMA 2 7B). Mixing a small portion of general text has limited effect on reducing the learned biases.

**Mixing unbiased data reduces learned bias, but only to a limited degree.** Mixing general corpus with domain corpus is a common strategy to avoid forgetting and over-adaptation. We examine the effect of data mixture using likelihood ratios in Figure 8, mixing Wikipedia text[1] into the PubMed corpus. The results indicate that mixing a small portion of general text reduces the learned biases, but the reduction is limited and only increases modestly with the proportion of general text. This shows that while it is possible to reduce the learned biases without affecting knowledge learning by mixing a small portion of general corpus, eliminating or considerably attenuating the biases may require a high general-to-domain data ratio, making it very uneconomic in terms of training cost.

## 4 RELATED WORK

**Dataset bias and shortcut learning.** Datasets used in machine learning often inevitably contain biases in the data distribution (Torralba & Efros, 2011). These superficial correlations in the data can be learned as a shortcut to achieve good performance on the training set (Geirhos et al., 2020). The issue is more pronounced in neural networks due to the tendency to learn simple features first, a phenomenon known as spectral bias (Rahaman et al., 2019; Xu et al., 2019). By adapting to biases, language models can achieve loss reduction without learning much underlying knowledge. Such "surface learning" (Geirhos et al., 2020) is analogous to the "principle of least effort" in linguistics where language speakers generally try to minimize effort in communication (Chang, 2016).

**Continual learning in language models.** Finetuning pre-trained language models can improve model's performance on a new domain (Chen et al., 2021; Lewkowycz et al., 2022) or a series of domains via continual pre-training (Gupta et al., 2023; Jin et al., 2022; Ke et al., 2023). Jang et al. (2022) specifically study knowledge learning in continual pre-training. Forgetting is frequently observed in continual pre-training and all the above work implement techniques to alleviate forgetting. Rehearsal (Chaudhry et al., 2019), regularization (Kirkpatrick et al., 2017), parameter isolation (Rusu et al., 2016), or a combination of multiple methods are often used. Mixing general corpus into the finetuning data also serves as a particular form of rehearsal.

**Data generation with LLMs.** LLMs such as GPT-3 (Brown et al., 2020) have been used to label examples for a variety of tasks (Liang et al., 2021; Hsieh et al., 2023). The generated labels can be used to train smaller specialized models as a form of knowledge distillation (Hinton et al., 2015). LLMs have also been used to generate rationales and reasoning steps, enabling the transfer of reasoning abilities (Fu et al., 2023; Ho et al., 2023; Li et al., 2023). They also generates instruction data for instruction-tuning and alignment of LMs (Wang et al., 2023b; Honovich et al., 2023).

**Decomposition analysis of text.** Separating the content and form has been a traditional approach in literary theories Eagleton (2011). Content analysis includes aspects like themes, ideas, and the narrative, while form (style) analysis deals with the use of literary devices like metaphors, tones, and the organization of the text. Mutiple linguistic theories further decompose the content of text into an overall topic and specific information, for example topic-focus articulation (Sgall et al., 1986) and theme-rheme analysis (Halliday, 1994).

---

[1]20230901 dump from `https://dumps.wikimedia.org`, English only

In machine learning, the three components constitute invididual topics of study. For example, topic modeling (Hofmann, 1999; Blei et al., 2003) studies the topic distribution of text, style transfer studies manipulation of style (Shen et al., 2017) and how to separate style from content (Fu et al., 2018). Information extraction (Brin, 1998; Banko et al., 2007) studies identifying factual information from text. Also, in document modeling, several work uses a hierachical structure to model the overall theme and specific information in text(Lin et al., 2015; Li et al., 2015; Nawrot et al., 2022), in a similar spirit as we did in this work.

## 5 DISCUSSION

Domain corpus used in language model finetuning can often exhibit significant homogeneity in topic and style, creating a statistically simple and salient feature easily learned by the model (Rahaman et al., 2019). We show that such adaptation creates a strong bias in the language model towards the training distribution. The bias stably increases with the amount of training data and can overshadow the learning of knowledge. The presence of a strong bias potentially makes the evaluation of knowledge learning more difficult, as an overly strong bias might interfere with the general reasoning abilities of the model.

Our observation shows that the topic and style biases are learned quickly and require little model capacity, which could mean that bias learning is hard to avoid in domain finetuning. Mixing general corpus in the finetuning data reduces the bias but adds significant training cost. This also poses a challenge for lifelong learning of language models. For example, when LLMs are used as general purpose agents, we want them to learn new knowledge from data without adapting too much to any individual data distribution.

While the current study aim to uncover the learning dynamics in domain finetuning, we believe that by identifying bias learning as a major hindrance in domain finetuning and showing the distinct behaviors of bias learning and knowledge learning, we also pointed out potential directions to improve knowledge learning and forgetting mitigation. For example, based on the observation that bias learning mostly happens on the first few tokens of each sequence, we could mask out the loss from the first few tokens in the finetuning objective to expect reduced bias learning. Based on the different capacity requirement of bias and knowledge learning, in principle we could use a small low-rank adapter to learn the bias, and subtract its weights from the full finetuned model to remove the bias while keeping the learned knowledge. We leave the exploration of such methods for future work.

**Limitations.** While our analysis leads to interesting findings on the learning dynamics of language models, it is limited in the following ways:

- Separability of text-generating factors: the decomposition of generating factors is only approximate and there may not be a generally agreed way to decompose. The boundary between content and style is not always clear, for example, terminology use is part of content and is also part of style. Interdependence between content and style sometimes prevents changing one factor without changing the other. In most domain corpora, the separation is clear enough for our analysis, evidenced by the quality of rewritten documents.

- Quality of rewriting with LLMs: several issues may limit the quality of generated rewrites. Safety alignment: requests for rewriting in the biomedical domain are sometimes rejected by LLM due to safety alignment to reduce harmful outputs (Kenton et al., 2021). Pretraining bias: LLMs may tend to generate text with certain topics or styles under a general instruction, which may create a bias in the generated data. Hallucination (Ji et al., 2023): LLMs have a certain probability of generating factually inaccurate content.

- Data dependency: the quantitative observations would reflect certain characteristics of the corpus (for example, the style adaptation in training depends on the style distribution in the corpus). We compare with more domain corpus in Appendix D and found our qualitative observations generalizes to other domains.

- Limited training: we only finetuned on a maximum of 1 million documents (<1B total tokens). Although we observe a consistent trend of learning of different factors with the amount of training data, it may not generalize to very large training sizes.

ACKNOWLEDGMENTS

The work is supported by National Key R&D Program of China (2021ZD0113402). We thank the anonymous reviewers for helpful comments and feedback.

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

# A  DATA

## A.1  DATASET GENERATION

Instructions used for each generating each derived datasets are listed below. A non-cherry picked example is also provided.

We found that ChatGPT (GPT-3.5-turbo) is capable enough for this task with some tuning of the prompts. We also found that GPT-4 require significantly less tuning of the prompts to produce successful rewrites, and can produce successful rewrites for examples on which ChatGPT fails. We use ChatGPT (GPT-3.5-turbo) with the following prompts for generating the derived datasets for the analysis.

We use nucleus sampling with p=0.9 to reduce the likelihood of generating low-quality rewrites.

**PubMed-nonbiomedical**

Prompt:

*Please rewrite the following passage by changing its topic to an irrelevant topic in art, finance, education or software.*
*- please keep the structure of the passage strictly unchanged*
*- please directly give the output without any comments*

**PubMed-counterfactual**

Prompt:

*Please change the biomedical terms in the following passage into other random biomedical terms so that the biomedical knowledge is disrupted.*
*- please keep the main topic and the words that are not biomedical terms unchanged*
*- please directly give the output without any comments*

**PubMed-casual**

Prompt:

*Please rewrite the following passage using a casual style.*
*- please keep the content (including all terminology) strictly unchanged*
*- please directly give the output without any comments*

**PubMed-rap**

Prompt:

*Please rewrite the following passage using the style of rap.*
*- please keep the content (including all terminology) strictly unchanged*
*- please directly give the output without any comments*

**C4-biomedical**

Prompt:

*Please rewrite the following passage by replacing its main topic with a biology or medicine related topic (for instance, some disease, diagnosis, drug, or treatment).*
*- please keep the style and the structure of the passage unchanged*
*- please directly give the output without any comments*

**C4-counterfactual**

Prompt:

*Please swap all the nouns in the following passage into random related nouns so that every piece of information given become random and completely different from the original.*
*- every piece of information must be changed*
*- please directly give the output without any comments*

**C4-academic**

Prompt:

*Please rewrite the following passage using the style of an abstract of a research paper (without title).*
*- please keep all the content strictly unchanged*
*- please directly give the output without any comments*

We randomly sampled 1000 documents from PubMed and C4 (the validation split) respectively (each document have at least 500 characters), and generated derived datasets from the samples. The generated datasets are listed in Table 1.

## A.2 EXAMPLES OF GENERATED TEXT

### PubMed (original)

*According to preliminary data, seroconversion after mRNA SARS-CoV-2 vaccination might be unsatisfactory in Kidney Transplant Recipients (KTRs). However, it is unknown if seronegative patients develop at least a cellular response that could offer a certain grade of protection against SARS-CoV-2. To answer this question, we prospectively studied 148 recipients of either kidney (133) or kidney-pancreas (15) grafts with assessment of IgM/IgG spike (S) antibodies and ELISpot against the nucleocapside (N) and the S protein at baseline and 2 weeks after receiving the second dose of the mRNA-1273 (Moderna) vaccine. At baseline, 31 patients (20.9%) had either IgM/IgG or ELISpot positivity and were considered to be SARS-CoV-2-pre-immunized, while 117 (79.1%) patients had no signs of either cellular or humoral response and were considered SARS-CoV-2-naïve. After vaccination, naïve patients who developed either humoral or cellular response were finally 65.0%, of which 29.9% developed either IgG or IgM and 35.0% S-ELISpot positivity. Factors associated with vaccine unresponsiveness were diabetes and treatment with antithymocytes globulins during the last year. Side effects were consistent with that of the pivotal trial and no DSAs developed after vaccination. In conclusion, mRNA-1273 SARS-CoV-2 vaccine elicits either cellular or humoral response in almost two thirds of KTRs.*

### PubMed-nonbiomedical

*According to preliminary data, the use of grayscale in abstract painting might be unsatisfactory in terms of conveying emotions. However, it is unknown if paintings created without grayscale can still evoke a certain level of emotional response. To answer this question, we conducted a prospective study involving 148 artists who work primarily with either acrylic (133) or oil (15) paints, assessing the use of grayscale and color saturation in their artworks. The assessment was done at baseline and 2 weeks after the artists experimented with incorporating grayscale into their paintings. At baseline, 31 artists (20.9%) had already used grayscale or experimented with it in their artworks, and were considered pre-immunized to grayscale, while 117 (79.1%) artists had no signs of using grayscale and were considered grayscale-naïve. After the experiment, grayscale-naïve artists who incorporated grayscale into their paintings accounted for 65.0% of the participants, with 29.9% using grayscale as part of their color palette and 35.0% achieving grayscale effects through techniques such as cross-hatching. Factors associated with unresponsiveness to grayscale experimentation were the artists' preference for vibrant colors and their training in traditional painting techniques. Overall, the results indicate that the use of grayscale in abstract paintings can elicit a response in almost two thirds of artists.*

### PubMed-counterfactual

*According to preliminary data, seroconversion after DNA HIV-1 vaccination might be unsatisfactory in Brain Tumor Recipients (BTRs). However, it is unknown if seronegative patients develop at least a cellular response that could offer a certain grade of protection against HIV-1. To answer this question, we prospectively studied 148 recipients of either brain (133) or brain-pancreas (15) grafts with assessment of IgM/IgG spike (S) antibodies and ELISpot against the nucleocapside (N) and the S protein at baseline and 2 weeks after receiving the second dose of the DNA-7532 (GenBio) vaccine. At baseline, 31 patients (20.9%) had either IgM/IgG or ELISpot positivity and were considered to be HIV-1-pre-immunized, while 117 (79.1%) patients had no signs of either cellular or humoral response and were considered HIV-1-naïve. After vaccination, naïve patients who developed either humoral or cellular response were finally 65.0%, of which 29.9% developed either IgG or IgM and 35.0% S-ELISpot positivity. Factors associated with vaccine unresponsiveness were diabetes and treatment with antithymocytes globulins during the last year. Side effects were consistent with that of the pivotal trial and no DSAs developed after vaccination. In conclusion, DNA-7532 HIV-1 vaccine elicits either cellular or humoral response in almost two thirds of BTRs.*

### PubMed-casual

*So, here's the deal: the early data suggests that people who have undergone kidney transplants (yeah, those Kidney Transplant Recipients, or KTRs) might not be getting a good immune response after getting the mRNA SARS-CoV-2 vaccine. But here's the thing we don't know yet: even if these patients don't have antibodies, could they still have some sort of cellular response that gives them at least some level of protection against SARS-CoV-2? To find out, we did a study with 148 people who got either a kidney (133 of them) or a kidney-pancreas (15 of them) transplant. We checked their IgM/IgG spike (S) antibodies and did ELISpot tests to see their response to the nucleocapside (N) and S proteins before the vaccine and two weeks after they got the second dose of the mRNA-1273 (Moderna) vaccine. At the start, we found that 31 patients (that's about 20.9% of them) already had either IgM/IgG antibodies or a positive ELISpot result, meaning they had some level of pre-existing immunity to SARS-CoV-2. The other 117 patients (about 79.1% of them) showed no signs of either a cellular or humoral response and were considered SARS-CoV-2-naïve. After getting vaccinated, we found that 65.0% of the initially naïve patients ended up developing either a humoral or a cellular response. Out of those, 29.9% had either IgG or IgM antibodies, and 35.0% had a positive S-ELISpot result. We also looked at some factors that might be linked to not responding well to the vaccine, and found that diabetes and treatment with antithymocyte globulins in the past year were associated with a lower response. The side effects were pretty much what we expected based on the main trial, and we didn't see any development of donor-specific antibodies after vaccination. So, to wrap it up, the mRNA-1273 SARS-CoV-2 vaccine seems to trigger either a cellular or a humoral response in almost two-thirds of kidney transplant recipients.*

## PubMed-rap

*Yo, check it out, here's the deal,*
*Preliminary data reveals what's real,*
*Seroconversion after SARS-CoV-2 vax,*
*Might not be satisfactory in KTRs, that's the facts.*
*But hold up, can seronegative patients shine?*
*With some cellular response to draw the line,*
*Against SARS-CoV-2, can they protect?*
*That's the question we're here to dissect.*
*Prospective study, 148 recipients in sight,*
*Kidney or kidney-pancreas, they're in the fight,*
*Testing IgM/IgG spike antibodies, so fly,*
*ELISpot against N and S, let's give it a try.*
*At baseline, 31 patients were pre-immunized,*
*With IgM/IgG or ELISpot, they were baptized,*
*But 117 patients showed no response at all,*
*SARS-CoV-2-naïve, they stood tall.*
*After vaccination, the game changed,*
*65.0% of the naïve were rearranged,*
*With humoral or cellular response in their veins,*
*29.9% with IgG/IgM, the victory remains.*
*And 35.0% with S-ELISpot positivity,*
*They're building defenses with ferocity,*
*Factors linked to unresponsiveness, it's true,*
*Diabetes and antithymocyte treatment can skew.*
*Side effects aligned with the pivotal trial,*
*No DSAs developed, that's a smile,*
*In conclusion, the mRNA-1273 vaccine,*
*Brings cellular or humoral response on the scene.*
*KTRs, listen up, almost two-thirds rejoice,*
*With immunity rising, you have a choice,*
*To fight off SARS-CoV-2, stand tall and bold,*
*Thanks to Moderna's vaccine, the story unfolds.*

### A.3 QUALITY ASSESSMENT OF GENERATED TEXT

**Quality.** The per-token perplexity of the original and derived datasets under LLaMA 2 13B is listed in Table 2. We can see that the perplexity of the derived datasets are higher than the original datasets but not much (except for the counterfactual rewrites which produce text contradicting common knowledge, therefore can have high perplexity under a language model). The generated

text are of acceptable quality and are not significantly different from the original text in terms of natural-ness judged by a language model.

| Dataset | Perplexity |
|---|---|
| *Original datasets* | |
| PubMed | 5.66 |
| C4 | 6.68 |
| *Derived datasets* | |
| PubMed (counterfactual) | 9.29 |
| PubMed (nonbiomedical) | 9.04 |
| PubMed (casual) | 6.20 |
| PubMed (rap) | 8.13 |
| C4 (counterfactual) | 13.03 |
| C4 (biomedical) | 7.38 |
| C4 (academic) | 6.80 |

Table 2: Perplexity of original and derived datasets under LLaMA 2 13B.

**Adherence.** We also evaluate the adherence of the generated text to the instruction. We randomly sampled 100 examples from each derived dataset and asked human annotators to label whether the generated text successfully comply to the instruction. "Good" means the generated text generally comply to the instruction, "partial" means the generated text only comply to part of the instruction (for example, the text was successfully changed to the requested style but the content was also significantly changed), and "bad" means the generated text does not comply to the instruction. The results are listed in Table 3. We can see that the generated text are generally of high adherence to the instruction. It seems that changing style has a higher success rate than changing content, which could mean that identifying and changing style is easier for a LLMs.

| Dataset | Good | Partial | Bad |
|---|---|---|---|
| PubMed (counterfactual) | 93 | 6 | 1 |
| PubMed (nonbiomedical) | 97 | 2 | 1 |
| PubMed (casual) | 100 | 0 | 0 |
| PubMed (rap) | 100 | 0 | 0 |
| C4 (counterfactual) | 97 | 3 | 0 |
| C4 (biomedical) | 94 | 5 | 1 |
| C4 (academic) | 96 | 3 | 1 |

Table 3: Adherence of generated text to the instruction.

# B  FINETUNING SETUP

To determine the learning rate, we perfomed grid search and train models under 100k documents with learning rates {1e-6, 3e-6, 1e-5, 3e-5, 1e-4, 3e-4} for full-finetune and {1e-5, 3e-5, 1e-4, 3e-4, 1e-3, 3e-3} for low-rank finetune. The final learning rate is chosen as the one that gives the lowest perplexity on the validation set. We use a batch size of 64.

During finetuning, we treat each document as a separate sequence to keep the structure of the document for better analysis of per-position likelihoods. This is different from a common LM pre-training setup where all the documents are concatenated together. All documents are truncated to a maximum of 1024 tokens.

Finetuning is performed with Huggingface's transformer library (Wolf et al., 2020), with bfloat16 mix-precision on NVIDIA A100 GPUs.

## C  RESULTS ON MORE DOMAIN DATA

To explore the effect of finetuning on other domain data, we perform similar analysis on the legal domain and the customer review domain with the setup in Section 3.1. For the legal domain, we use the Pile of Law corpus (Henderson et al., 2022). We finetune LLaMA 2 7B on up to 1M court opinions from the "Court Listener Opinions" subset of the Pile of Law corpus. For the customer review domain, we use the Amazon reviews dataset (Ni et al., 2019). We finetune LLaMA 2 7B on up to 1M reviews of automotive products from the "automotive" subset of the Amazon reviews dataset. We also rewrite the original documents to generate derived datasets. The datasets used are listed in Table 4.

| Dataset | Source | Topic | Factuals | Style |
|---|---|---|---|---|
| *Original datasets* | | | | |
| Pile of Law | - | legal | factual | court opinion |
| Amazon reviews | - | automotive | factual | customer review |
| *Derived datasets* | | | | |
| Pile of Law (nonlegal) | Pile of Law | **nonlegal** | factual | court opinion |
| Pile of Law (counterfactual) | Pile of Law | legal | **counterfactual** | court opinion |
| Pile of Law (casual) | Pile of Law | legal | factual | **casual** |
| Pile of Law (rap) | Pile of Law | legal | factual | **rap** |
| Amazon reviews (nonautomotive) | Amazon reviews | **nonautomotive** | factual | customer review |
| Amazon reviews (counterfactual) | Amazon reviews | automotive | **counterfactual** | customer review |
| Amazon reviews (academic) | Amazon reviews | automotive | factual | **academic** |
| Amazon reviews (rap) | Amazon reviews | automotive | factual | **rap** |

Table 4: Datasets used for probing language models on the legal and the customer review domain. Refer to Table 1 for explanation of the columns.

Figure 9 and 10 shows the change of likelihood ratios between different topics, styles and factual during finetuning on the legal and the customer review domain. Similar to the biomedical domain, the likelihood of the dominant topic and style in the training corpus increases significantly during finetuning with respect to other topics and styles. The Pile of Law data has a dominant topic of legal affairs and a dominant style of court opinion. The Amazon reviews data has a dominant topic of automotive products but the style is generally casual and more diverse than other domain text, therefore the adaptation of style prior is less significant. This shows that the presentation of dominant topic and style in the training corpus would invariably leads to strong adaptation of topic and style priors under the present language model finetuning regime.

For all the domains, the factual/counterfactual likelihood ratio changes at a significantly slower rate than the topic and style likelihood ratios, showing that the effect of topic and style adaptation on text modeling probabilities are much more significant than the effect of knowledge learning.

## D  ADDITIONAL LM EVALUATION

**Language model evaluation: general abilities**  We evaluate LLaMA 2 7B finetuned in our analysis on Hellaswag (Zellers et al., 2019), ARC (Challenge Set) (Clark et al., 2018) and MMLU (Hendrycks et al., 2021), using the Language Model Evaluation Harness framework Gao et al. (2021). Zero-shot and 5-shot performance is presented in Table 5.

| Learning rate | Training size | Hellaswag | | ARC-Challenge | | MMLU | |
|---|---|---|---|---|---|---|---|
| | | 0-shot | 5-shot | 0-shot | 5-shot | 0-shot | 5-shot |
| Baseline | - | 76.0 | **78.1** | **46.2** | **53.2** | **42.6** | **46.6** |
| 3e-6 | 1M | **76.5** | 77.8 | 46.0 | 52.5 | 42.2 | 46.5 |
| 1e-5 | 1M | 75.7 | 76.9 | 44.4 | 50.0 | 40.5 | 45.0 |
| 2e-5 | 1M | 73.9 | 75.2 | 41.7 | 46.1 | 37.3 | 42.1 |

Table 5: Evaluation of LLaMA 2 7B finetuned on PubMed.

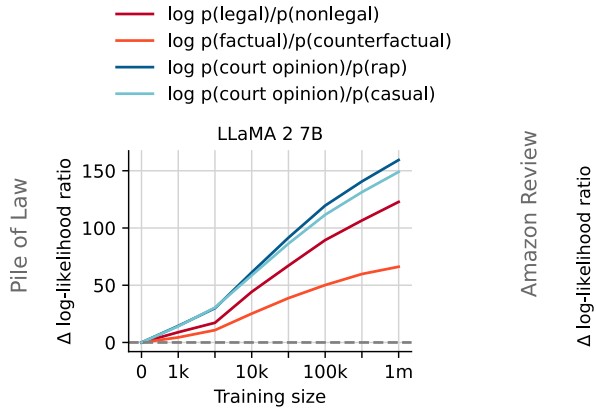
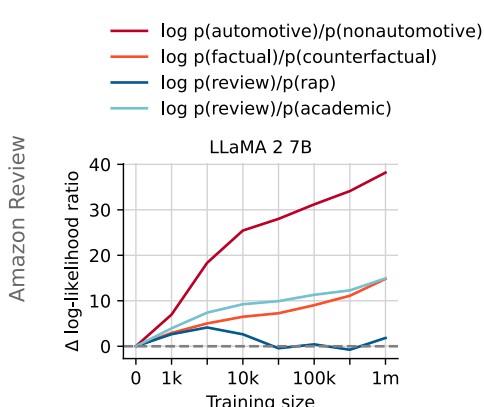

Figure 9: Change of likelihood ratios of content and style factors with the amount of training on Pile of Law.

Figure 10: Change of likelihood ratios of content and style factors with the amount of training on Amazon reviews.

**Language model evaluation: medical knowledge** To verify that medical knowledge is learned through finetuning on the PubMed corpus, we evaluate LLaMA 2 7B on clinical subsets from MMLU, following the Med-PaLM 2 paper (Singhal et al., 2023). Results (5-shot) are listed in Table 6. We further plot the factual/counterfactual likelihood ratio and the average accuracy on MMLU clinical subsets on the same graph in Figure 11. The two curves show a similar trend, indicating that the factual/counterfactual likelihood ratio is indeed an indicator of the learning of biomedical knowledge, to the degree that MMLU clinical subsets reflects knowledge learning on PubMed.

| Training size | Anatomy | Clinical knowledge | College biology | College medicine | Medical genetics | Professional medicine |
|---|---|---|---|---|---|---|
| Baseline | 46.7 | 45.7 | 46.5 | 41.0 | 51.0 | 51.8 |
| 1K | 45.2 | 46.0 | 46.5 | 42.2 | 52.0 | 51.8 |
| 10K | 45.9 | 46.0 | 45.1 | 42.2 | 53.0 | 52.6 |
| 100K | 47.4 | 46.4 | 46.5 | **43.4** | 52.0 | 52.9 |
| 1M | **48.1** | 46.4 | 46.5 | 42.8 | **54.0** | **54.4** |

Table 6: Evaluation of LLaMA 2 7B finetuned on PubMed.

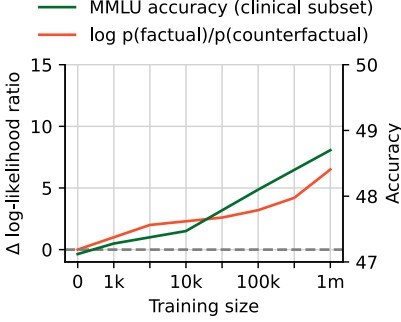

Figure 11: Comparing the change in the factual/counterfactual likelihood ratio and question answering accuracy on MMLU clinical subsets.

