# OpenReview forum: "Dissecting learning and forgetting in language model finetuning"
_ICLR.cc/2024/Conference — ICLR 2024 poster_

### Official Review · Reviewer_o2MC · 2023-10-28

**Soundness:** 3 good
**Presentation:** 3 good
**Contribution:** 3 good
**Rating:** 8
**Confidence:** 4

**Summary:**

The paper studies the effects of finetuning on an LM. The authors disentangle the effects of the finetuning along three “dimensions”: topic, style, and factual knowledge. To achieve that, they leverage ChatGPT to generate a series of texts that are only different in one of those facets. Provided such texts, one can estimate log-likelihood ratio of different styles (for instance), by calculating differences of cross-entropies of the model on the texts.
The generated texts were verified by human judgements.

The experimental study is performed using two corpora (BioMed and C4) and three LMs (GPT-2 XL, LLaMa 2 - 7B & 13B).

The paper reports the following findings: (a) topic and style changing rapidly,(b) topic and style biases are independent, (c) topic and style require minimal capacity to be learned, in contrast to knowledge, (d) mixing in unbiased data only reduces the biases to a certain degree.

**Strengths:**

* I find the topic of the investigation quite novel. I believe that the approach taken is original and innovative, in particular building a corpus that allows disentangling style/topic/factual knowledge. I also like the way LoRa was used to measure the capacity required for learning different facets.
* The authors are sharing the data and code.
* The reported experiments have provided some applicable insights, e.g. wrt the data mixing.

**Weaknesses:**

* Using synthetic data, generated by ChatGPT, might introduce some hidden biases. It is not given that the same findings could be found if we had natural data.
* It is not clear if the same approach can be generalized to any other characteristics?

Typos:
* “by just changing the order of decomposition in 1” -> “...in Eq. 1”

**Questions:**

* I wonder if there are other factorizations which can be studied in the same setup, apart from style/topic/knowldge?

---

> ### Author Response · Authors · 2023-11-19
> **Response to the reviewer**
>
> We would like to thank the reviewer for the insightful comments and suggestions. We have revised the paper incorporating many of the suggestions. Please let us explain our responses to the reviewer's comments below:
>
>
> **Hidden biases of synthetic data**
>
> We agree with the reviewer that text generated with ChatGPT can have hidden biases, and we mentioned some forms of possible bias in the limitations of the paper. It would make the results of our paper more robust if we had natural data, but unfortunately the data we require for controlled-variable analysis (e.g., two document having the same content but different styles) is unlikely to be available in natural corpora. The advantage of using synthetic data is precise control on the components on text, so that we can have good precision in locating the variables we want to study which increases the reliability of our results.
>
> **Generalization to other characteristics of text**
>
> The decomposition of text into generating factors is not unique, as long as the decomposition of $p(x)$ into conditional probabilities corresponds to a valid model of text-generation process. For example, we could decompose text into a semantic part and a syntactic part, suggested by one of the reviewers. Such a decomposition would corresponds to a model of text generation where the semantic information is first determined, then the syntax of the document is determined based on the semantic information.
>
> One would find more ways to decompose text for studying its various characteristics from linguistic theories. For example, stylistics separates the style of text into linguistic features such as vocabulary, syntax, and figurative language. Narratology separates the narrative of text into components like plot, characters, and narrative perspective. For specific types of text, we can define more components to analysis than general text.
>
> As long as the separation of one characteristic from the rest of the text is clear enough and meaningful, we could apply our dissection analysis method to analyze how the characteristic is learned in finetuning. More precisely, "separation" means that the chosen factor has a degree of freedom that is independent of other factors, so that modifying it while keeping other factors still result in valid and coherent text. For example, the plot in a narrative text can be changed while keeping the characters and narrative perspective the same.
>
>
> **Typo in the paper**
>
> We thank the reviewer for pointing out the typo and we have corrected it in the revised paper.
>
> **Generalization to other factorizations**
>
> We answered this question in combination with the previous question and provided our response above.
>
>
> **Other paper updates**
>
> We also kindly ask the reviewer to refer to the "comments to all reviewers" section at the top of this page for a list of newly added contents in the revised paper. We included theoretical support of our methods, new domain corpora and more knowledge evaluation tasks to generalize our findings. We hope that the revised paper is getting even more clear and complete with the help of the reviewers.

---

### Official Review · Reviewer_QBAp · 2023-10-29

**Soundness:** 2 fair
**Presentation:** 2 fair
**Contribution:** 2 fair
**Rating:** 5
**Confidence:** 3

**Summary:**

The paper investigates the impact of finetuning language models on domain-specific texts and how it affects their general performance. The authors show that finetuning alters the model's preferences for topics and styles significantly, learning these features quickly and with minimal capacity. Factual knowledge, however, is acquired more slowly and requires greater capacity. The study's insights into language model learning dynamics could guide future enhancements in domain adaptation and help address the challenge of model forgetting during continuous learning.

In this study, the authors fine-tuned three models in increasing scales (GPT2-XL, LLaMa2 7B, and 13B) on PubMed abstracts with different scales of datasets up to 1M abstracts.

**Strengths:**

1. Investigating the changes language models undergo after finetuning continues to be a highly relevant and evolving area of study, despite prior coverage in academic literature.

2. The research offers key empirical insights into the differential impact of finetuning on language models, revealing a more pronounced effect on style and topic preferences compared to factual knowledge. These findings enhance our understanding of language model training dynamics and are instrumental in formulating more effective training methodologies.

3. The researchers conducted extensive experiments on three language models of considerable size, particularly from an academic perspective.

**Weaknesses:**

* The assertion that each prediction by a language model can be broken down into components of writing style, topic, and factual knowledge requires further justification or explanation. The paper should present a stronger argument or provide additional evidence to substantiate this claim.

* The primary message or conclusion of the paper is ambiguous. The authors need to clarify the central thesis to ensure that readers can grasp the main contribution of the work. what is the takeaway from this research?

* While the prose is generally lucid, the paper's structure, particularly the introduction, could use refinement to enhance its readability and impact.

For improved clarity and presentation, the following suggestions are offered:
1. The introduction would benefit from concrete examples illustrating the key domains of style, topic, and factual knowledge to help contextualize the subsequent findings.
2. Details regarding the fine-tuning process are scant. Clarification of which specific models are assessed and the precise nature of the fine-tuning would provide a more robust understanding of the study's scope.
3. The transition from discussing fine-tuning effects to probing methods in the third paragraph of the introduction is somewhat abrupt. A smoother segue that connects these topics would aid reader comprehension.
4. In the fourth paragraph, where style and topic biases are introduced, it would be helpful to include examples or elaborate on what these biases entail to furnish readers with a clearer picture of these concepts.

**Questions:**

* The basis of the method assumes that p(x)=p(topic,style,factual), but is there a justification to that decomposition? what about arithmetic? how does it fall to this decomposition?

---

> ### Author Response · Authors · 2023-11-19
> **Response to the reviewer**
>
> We would like to thank the reviewer for the insightful comments and suggestions. We have revised the paper incorporating many of the suggestions. Please let us explain our responses to the reviewer's comments below:
>
>
> **Decomposition of text-generating factors**
>
> We could find theories from linguistics and literary analysis that endorse the approach of decomposition of text into content and style. Separating the content (what is being said) and form (how it is being said) has been a traditional approach in literary theories (Eagleton, 2011). Content analysis deals with themes and the narrative while form (style) analysis studies the use of literary devices like metaphors and the organization of the text.
>
> Some linguistics theories further support decomposing the content into an overall topic and specific, detatiled information. For example, topic-focus articulation (Sgall et al., 1986) distinguishes between the "topic" of a sentence (what the sentence is about, or its theme) and the "focus" (new or important information in the sentence). Theme-rheme analysis (Halliday, 1994) divides a sentence into "theme" (the departure point of the clause, what it's about) and "rheme" (the rest of the clause, what is being said about the theme). It can also be extended to larger text structures, where the overall theme of the text is distinguished from specific, detailed information.
>
> In machine learning and NLP, there are many work that study the three aspects separately. For example, topic modeling (Hofmann, 1999; Blei et al., 2003) studies the topic distribution of text in a corpus, style transfer studies manipulation of the style of text (Shen et al., 2017; Fu et al., 2018) (and also how to separate style from content (Fu et al., 2018)), and information extraction (Brin, 1998; Banko et al., 2007) studies the identification of factual information from text. Also, in document modeling, several work uses a hierachical structure to separately model the overall theme and specific information (Lin et al., 2015;
> Li et al., 2015; Nawrot et al., 2022), in a similar spirit as we did in this work.
>
> The citation and discussion on the above mentioned work can be found in the Related Work section of the revised paper. We hope that we provided sufficient evidence from past work for justification of the decomposition used in our approach.
>
> **A summary of contributions with better focus**
>
> We have rewritten the summary of contributions in the introduction section to better highlight the main contributions of the paper. The main contributions of our paper are the following two observations:
>
> * Domain finetuning leads to a significant change in the topic and style priors of the language model, biasing them towards the training data. Effect caused by such bias dominates the learning and forgetting observed in finetuning.
>
> * Topic and style biases are learned like simple features, while factual knowledge are learned like complex features in finetuning. We present the significant differences between their learning dynamics in multiple aspects.
>
> The details of each observation are succinctly summarized under the two bullet points at the end of the introduction section. We hope that the revised introduction can help readers better grasp the main contributions of the paper.

---

> > ### Author Response · Authors · 2023-11-19
> > **(continued)**
> >
> > **An rewritten introduction with improved clarity**
> >
> > We have significantly rewritten the introduction section to improve the following aspects suggested by the reviewer:
> >
> > * In introducing the three text-generating factors, we explained the meaning of style, topic, and factual knowledge factors, and uses the current paper as an example to illustrate what each factor corresponds to:
> >
> >   - style: the structure, tone and diction, e.g., academic writing in ICLR paper format;
> >
> >   - topic: the overall theme, e.g., "language model finetuning";
> >
> >   - factual knowledge: the detailed factual information, e.g., methods, citation, and results in this paper.
> >
> > * We introduced the specific models assessed (open LMs like LLaMA) and the nature of finetuning (causal language modeling) early in the introduction.
> >
> > * We rewrite the paragraph that introduces probing methods, so that we introduce our probing method as a natural choice given prior related work and our goal of studying learning dynamics. We think now it is much smoother and no longer abrupt.
> >
> > * We explained that the style and topic biases means that the during generation, the language model is biased towards generating text with the style and topic present in the training data and less likely to generate text with other topics and styles.
> >
> > **Justification for the decomposition**
> >
> > We have included references to both linguistic theories and machine learning studies that support the decomposition of text into topic, style and factural knowledge in the Related Work section of the revised paper. For arithmetics content, for example in mathematical text, the topic could be the mathematical topic (e.g., calculus), the style be the writing style of mathematical text, and the factual knowledge being the mathematical facts (e.g., how the right hand-side relates to the left-hand side in an arithmetics expression).
> >
> > **Other revisions of the paper**
> >
> > We also kindly ask the reviewer to refer to the "comments to all reviewers" section at the top of this page for a list of newly added contents in the revised paper. We also included new domain corpora and more knowledge evaluation tasks to generalize our findings. We hope that the revised paper is getting even more clear and complete with the help of the reviewers.

---

### Official Review · Reviewer_xGn4 · 2023-10-30

**Soundness:** 3 good
**Presentation:** 2 fair
**Contribution:** 3 good
**Rating:** 5
**Confidence:** 4

**Summary:**

This paper dissects the effect of fine-tuning on learning and forgetting of language style, topic, and factual knowledge. The authors use instruction-following LLMs to automatically construct corpus with controlled factors above. The authors performed extensive analysis across different LM types and summarized several empirical findings, among which they show topic and style priors are easy to learn but factual knowledge is not.

**Strengths:**

- The method how the analysis is performed is novel. Creating training and evaluation corpora with controlled differences (topics, style, factual knowledge) by prompting instruction following LLMs is interesting and inspiring.
- The analysis is extensive and is performed under various configurations (like the choice of LM, size of the training corpora)
- The outcomes of analysis are interesting and relevant to future research that study lifelong learning of LMs.

**Weaknesses:**

- Although other configurations are very extensive, the choice of training and evaluation corpora and exclusively original or variants of PubMed and C4.
- The three text-generating factors (styles, topics, facts) may not always be clearly separable of extensive enough in every corpora. The authors discussed this limitation in their limitation section.
- Clarity issue: I feel the plots very hard to read because the captions are too generic and not self-contained. I suggest to briefly summarize the findings or implications in the captions.
- Clarity issue: some legends in plots such as Figure 4 are not explained in text (e.g. readers may be confused about "C4 -factuals" before they associate them with "C4-counterfactual" in Table 1)
- Though the authors pointed out the hardness of learning factual knowledge without learning style and topic bias, the authors' attempts failed to improve such performance at the end of Sec. 3. I suggest to provide some future directions about how the analysis will be beneficial the challenge of learning factual knowledge above.
- The authors focused on evaluation of LM loss throughout the paper. I think this is fine for style and topics, but factual knowledge, evaluating LM loss is not clean enough because only a few tokens in a sentence are related to facts. The authors could create cloze-style  or question answering evaluation sets that focus exclusively on generation of factual knowledge.

**Questions:**

- There is a "side note" in page 7: "When capacity is limited, the topic ratio and factual ratio simultaneously reduce on Pubmed in Figure 6." I did not see topic ratio reduces in Figure 6. Is this information supposed to be told by Figure 6?

---

> ### Author Response · Authors · 2023-11-19
> **Response to the reviewer**
>
> We would like to thank the reviewer for the insightful comments and suggestions. We have revised the paper incorporating many of the suggestions. Please let us explain our responses to the reviewer's comments below:
>
> **Corpora for evaluation**
>
> We included two new domains, the legal domain and the customer review domain, in our analysis to verify the generalization of our finding. For the legal domain, we use the "Court Listener Opinions" subset from the Pile of Law corpus. For the customer review domain, we use the "automotive" subset from the Amazon reviews dataset. We use the same procedure as in our main study to finetune LLaMa 2 7B model, generate derived datasets, and probe the model accordingly.
>
> We show (in Figure 9-10 of the revised paper) that similar to the biomedical domain, for the legal and the customer review domain, likelihood of the dominant topic and style in the corresponding training corpus also increases significantly during finetuning. For all the domains, the factual/counterfactual likelihood ratio changes at a significantly slower rate than the topic and style likelihood ratios, showing that the effect of topic and style adaptation on langauge modeling modeling are much more significant than the effect of knowledge learning. This shows that our main findings are likely general phenomena in domain finetuning.
>
> The above results and discussions are included in the Appendix D of the revised paper.
>
> **Separability of text-generating factors**
>
> We agree that the text-generating factors may not always be clearly separable in every corpora, and this poses a limitation on the scope where the dissection analysis can be directly applied. However, we believe that the general idea of decomposition of text into simple features (such as topic, style) and complex features (such as factual knowledge) is still useful in understanding the learning dynamics in domain finetuning even in cases where the decomposition cannot be explicitly performed.
>
> **Improving factual knowledge learning**
>
> While the current study mainly aim to uncover the learning dynamics in domain finetuning, we believe that by identifying bias learning as a potential hindrances in knowledge learning and showing the properties of bias learning, we also pointed out potential directions to improve knowledge learning. We can use the discovered differences between bias learning and knowledge learning to design methods that encourage knowledge learning. For example, based on the observation that bias learning mostly happens on the first few tokens of the sequence, we could mask out the loss on the first few tokens in the finetuning objective to considerably reduce bias learning. Based on the different capacity requirement of bias and knowledge learning, in principle we could use a small low-rank adapter to learn the bias, and subtract its weights from the full finetuned model to remove the bias while keeping the learned knowledge.
>
> We add a part of discussion on potential directions to improve knowledge learning in the conclusion section of the revised paper.

---

> > ### Author Response · Authors · 2023-11-19
> > **(continued)**
> >
> > **Evaluation of factual knowledge**
> >
> > We agree that the factual knowledge of language models is more clearly evaluated with question answering tasks. As there may not be existing question answering datasets that directly evaluates knowledge in the PubMed corpus, we use the clinical subsets from the MMLU benchmark to evaluate models' knowledge of medical concepts, following Google's Med-PaLM 2 paper. The following results show that the models finetuned on PubMed generally have better knowledge of medical concepts than the baseline model:
> >
> > | Training size | Anatomy | Clinical knowledge | College biology | College medicine | Medical genetics | Professional medicine |
> > |---------------|---------|--------------------|-----------------|------------------|------------------|-----------------------|
> > | Baseline      | 46.7    | 45.7               | 46.5            | 41.0             | 51.0             | 51.8                  |
> > | 1K            | 45.2    | 46.0               | 46.5            | 42.2             | 52.0             | 51.8                  |
> > | 10K           | 45.9    | 46.0               | 45.1            | 42.2             | 53.0             | 52.6                  |
> > | 100K          | 47.4    | 46.4               | 46.5            | **43.4**         | 52.0             | 52.9                  |
> > | 1M            | **48.1**| 46.4               | 46.5            | 42.8             | **54.0**         | **54.4**              |
> >
> > We also compared the factual/counterfactual likelihood ratio proposed in this paper and the average accuracy on MMLU clinical subsets on the same graph as a cross-validation of the two metrics (Figure 11 of the revised paper). The two curves show a similar trend, indicating that the factual/counterfactual likelihood ratio is indeed an indicator of the learning of biomedical knowledge.
> >
> > The above results and discussions are included in the Appendix E of the revised paper.
> >
> > **Improving clarity of the figures**
> >
> > We added summaries of the main findings of each Figure in their captions (Figures 2-8) to make them more self-contained. We also add an explanation of the legends of Figure 4 and 6 in their captions to aid comprehension.
> >
> >
> > **The side note in Figure 6**
> >
> > The side note on page 7 is intended to explain the phenomenon that the topic ratio (red line) reduces with decreasing model capacity on PubMed (the left plot of Figure 6). The topic ratio on C4 (the right plot of Figure 6) is not affected by model capacity. We use the side note to explain this minor difference.
> >
> > **Other revisions of the paper**
> >
> > We also kindly ask the reviewer to refer to the "comments to all reviewers" section at the top of this page for a list of newly added contents in the revised paper. We also included theoretical support of our methods and a rewritten introduction section with more clarity and focus. We hope that the revised paper is getting even more clear and complete with the help of the reviewers.

---

### Official Review · Reviewer_DT5M · 2023-10-31

**Soundness:** 3 good
**Presentation:** 3 good
**Contribution:** 2 fair
**Rating:** 5
**Confidence:** 4

**Summary:**

- This paper presents a detailed analysis of the effects of fine-tuning of large language models on domain-specific downstream tasks/datasets.
- In doing so, authors break down the probability distribution of a text into its fundamental factors i.e., topic, style and factual knowledge, and study the effects of fine-tuning on the probability distribution over these three factors.
- It has been shown that in the early cycles of fine-tuning, the language model easily captures the topic and style information of the underlying text data thus introducing learning bias, which ultimately leads to an increase in the forgetting of the previous knowledge. However, the model is able to capture the factual knowledge in the later cycles of fine-tuning and also requires significant model capacity as compared to the model capacity required for capturing topic and style information.
- Extensive experimental evaluation asserts the claims made by authors and opens a new research direction in continual learning research.

**Strengths:**

- Quality
	- The motivation is well-founded and the claims are sound.
	- Experimental analysis is very detailed and explanatory.
- Clarity
	- Paper is clearly presented and easy to follow.

**Weaknesses:**

- Quality
	- As the topic of a document can be determined by the factual knowledge it contains then it might be redundant to keep the topic as a relevant factor in the text generation process and only style and factual knowledge might suffice which then could directly align with the syntax and semantics of the underlying text respectively.
- Significance
	- This paper presents a detailed technical analysis of the fine-tuning process of a language model on domain-specific downstream tasks/datasets. However, the outcomes of the study conform with the expected outcomes of fine-tuning a model on domain-specific data and hence this paper misses to provide any significant gainful insight into the fine-tuning process due to the following reasons:
		- In the PubMed dataset, as academic style is present across all abstracts with different factual knowledge, it is expected that the model will readily adapt to the academic style first before capturing the diverse type of factual knowledge.
		- Just like in topic modeling, the topic of a document is a broad sentiment and can be easily determined using a set of keywords. So, it will be easy for the model to detect/understand the topic of a document before reading the whole document and capturing the factual knowledge inside it. Therefore, it is expected for the model to easily understand the topic and style factors before capturing the factual knowledge inside it.
	- I am keen to hear the response of the authors on this and hope that they can change my point of view.

**Questions:**

- The C4 dataset could possibly contain the documents written in the "academic" style although in a different domain. Similarly, the C4 dataset could also contain documents related to the biomedical domain although having different factual information. So, it is possible that the model is adapting fast to the academic style and biomedical domain topic because it has already seen them in the pretraining data, but the diverse factual information in the PubMed dataset is new for the model, and that is why model is possibly taking time to capture that knowledge. Have authors taken this into consideration in their analysis of the finetuning process?

---

> ### Author Response · Authors · 2023-11-19
> **Response to the reviewer**
>
> We would like to thank the reviewer for the insightful comments and suggestions. We have revised the paper incorporating many of the suggestions. Please let us explain our responses to the reviewer's comments below:
>
> **The necessity of separating the topic factor**
>
>  *Separating the topic factor reflects a natural generation process of text, and allows us to study the many properties of topic bias.*
>
> We agree that the topic of a document can be determined by the factual knowledge it contains. However, considering the natural generation process of text (i.e., how a paragraph is written), it is often the case that the overall topic is determined first, then the factual information to include in the paragraph is selected. Our decomposition of $p(x)$ into $p(topic) p(factual|topic)$ reflects such a natural generation process, where $p(topic)$ models how likely a topic is chosen in the first step, and $p(factual|topic)$ models how likely certain factual information is included once the topic is chosen. As the reviewer pointed out, the alternative decomposiiton of $p(x)$ into $p(factual) p(topic|factual)$ will not make sense becuase $p(topic|factual)$ is deterministic.
>
> There are linguistics theories that support the decomposition of text into an overall topic and specific, detatiled information. For example, topic-focus articulation (Sgall et al., 1986) distinguishes between the "topic" of a sentence (what the sentence is about, or its theme) and the "focus" (new or important information in the sentence). Theme-rheme analysis (Halliday, 1994) divides a sentence into "theme" (the departure point of the clause, what it's about) and  "rheme" (the rest of the clause, what is being said about the theme). In machine learning and NLP, there are also prior work that endorse a hierachical decomposition of paragraphs which fascilitate analysis of text at different levels of semantic granularity (Lin et al., 2015; Li et al., 2015; Nawrot et al., 2022). We have included a new subsection in the Related Work section of the revised paper to discuss the above mentioned work, for a better justification of the text decomposition method used in our approach. (The citation and discussion on the above mentioned work can be found in the Related Work section of the revised paper.)
>
> Separating the topic factor from factual knowledge also allows us to study the many differences of the learning dynamics between topic bias and factual knowledge, which constitudes one of the two main contributions of our paper. We show that the topic biases and factual knowledge are learned very differently, which also evidences that to a certain extent the language model internally models topics separately from factual knowledge. It might have been harder to gain these insights if the topic factor is not separated from factual knowledge in the analysis. We have also revised the introduction section to better highlight the main contributions of the paper.

---

> ### Author Response · Authors · 2023-11-19
> **(continued)**
>
> **Significance of our findings**
>
> *Our findings starts from a simple intuition, but leads to an unexpected result, detailed understanding of different components in domain finetuning, and wide implications for future work.*
>
> We agree with the reviewer that topic and style are very simple features of text. We pointed out that given existing study on spectral bias and shortcut learning, it is expected that topic and style are learned faster than factual knowledge. This is the starting point of our study.
>
> With a dissection analysis, we found that the topic and style biases not only learns *fast*, but the *degree* of adaptation of topic and style priors is very significant, much larger than the degree of adaptation to factual knowledge. We think this high degree of biasing towards certain topic and style is not quite expected, as one could reasonably speculate that because topic and style are so salient and easily determinable, the model can easily recognize them during inference and does not need to adopt a strongly biased prior.
>
> We showed that the implication of bias learning is that it affect how we interpret loss/perplexity changes in finetuning. Large reduce in perplexity does not necessarily mean significant learning of new domain knowledge as we would often assume. Large increase in perplexity on general corpus also does not equal to catastrophic forgetting of factual knowledge. We also need to take care of the bias in finetuned models in applications, as they could potentially be too heavily biased towards certain topic and style in generation which may negatively impact application in real-world scenarios.
>
> As our second main contribution of the paper, we showed that *there are many more different properties between bias learning and knowledge learning besides the learning order*. We showed that the two types of learning have different capacity requirements, different sensitivity to learning rate and data composition, and affect modeling probabilities at different token positions. Also the topic and style biases are learned independently of each other.
>  These results are not found in previous literature and we believe they are not obvious derivatives from the nature of domain finetuning.
>
> We also point out (in the revised version of the paper) how these new insights into the fine-tuning process can be useful and its implications for future directions. We can use the discovered differences between bias learning and knowledge learning to design methods that encourage knowledge learning and reduce forgetting. For example, based on the observation that bias learning mostly happens on the first few tokens of the sequence, we could mask out the loss on the first few tokens in the finetuning objective to considerably reduce bias learning. Based on the different capacity requirement of bias and knowledge learning, in principle we could use a small low-rank adapter to learn the bias, and subtract its weights from the full finetuned model to remove the bias while keeping the knowledge. We believe that more understanding of the detailed learning dynamics in domain finetuning under a systematic analysis is very much needed as a basis to explore more finetuning techniques beyond the conventional causal language modeling recipe.

---

> > ### Author Response · Authors · 2023-11-19
> > **(continued)**
> >
> > **The effect of old vs. new information in finetuning**
> >
> > We really appreciate the reviewer's point on the effect of old vs. new information on the speed of learning in finetuning. We agree that combining the fact that 1) LLMs pretrained on general internet corpora likely have knowledge of lots of different topics and styles and 2) topic and style are simple features of text thus are easy to learn, it would be easy for the model to adapt to the topic and style of the domain corpus.
> > We have included the discussion on the effect of old vs. new information in the revised paper, as a possible explanation of the order of learning between biases and factual knowledge. We note that, similar to the previous point, the effect of old vs. new information explans the order of learning but  not the many other differences between bias learning and knowledge learning that we discovered in our analysis.
> >
> > **Other revisions of the paper**
> >
> >  We also kindly ask the reviewer to refer to the "comments to all reviewers" section at the top of this page for a list of newly added contents in the revised paper. We included theoretical support of our methods, new domain corpora and more knowledge evaluation tasks to generalize our findings. We hope that the revised paper is getting even more clear and complete with the help of the reviewers.

---

### Author Response · Authors · 2023-11-19
**Comments to all reviewers**

We would like to thank all the reviewers and express our appreciation for all the insightful comments and suggestions.

We have uploaded a revision of the paper by incorporating many helpful suggestions from the reviewers. This revision mainly include the following aspects:

* More domain corpora for evaluation: we add two new text domains in our analysis: the legal domain and the customer review domain. Results show that our findings generalize across different domains, reflecting a common phenomena in domain finetuning. The description of experiment settings of the two new domains, the new results, and discussions are included in the Appendix D of the revised paper.

* Evaluating factual knowledge with question answering: we add evaluation of knowledge learning with quesition answering datasets, and shows with side-by-side comparison that both the question answering performance and the proposed factual/counterfactual likelihood ratio are useful indicators of knowledge learning. The description of datasets, results, and discussions are included in the Appendix E of the revised paper.

* Text decomposition analysis in the literature: we add reference to linguistic theories that endorse the decomposition of text into topic, style, and factual information. We also reference to machine learning studies on the three aspects of text. We discuss how previous literature support and justify the text decomposition used in our approach. The added contents are included in the Related Work section of the revised paper.

* Rewritten introduction with more clarity and focus: we have rewritten the introduction section to better introduce the text decomposition analysis method we proposed in the paper with examples.
The new introduction also has an improved flow and better highlights the main contributions of the paper.

* Improved presentation of figures: we add summaries of the main findings to each figure to make them more self-contained. We add more explanation of the legends to help the reader better understand the figures.

* Discussion on directions to improve knowledge learning and future work: we add a part of discussion on potential directions to improve knowledge learning in the final section of the revised paper.

We hope that the revised paper is more clear and complete with enhancements in theoretical justification and empirical verification. We thank the reviewers again for their work that helped us improve the paper.

---

### Meta-Review · Area_Chair_FYuR · 2023-12-11

**Metareview:**

Strengths
* Investigating the changes language models undergo after finetuning is very relevant to ICLR
* The approach seems novel
* The authors are sharing the data and code
* The paper is well written

Weaknesses
* Assumes a decomposition in style, topic, and factual knowledge
* Primary message or conclusion of the paper is ambiguous (lack of actionnable recommendations)

**Justification For Why Not Higher Score:**

See weaknesses.

**Justification For Why Not Lower Score:**

See strengths.

---

### Decision · Program_Chairs · 2024-01-16

Accept (poster)